# Assessment of a Community-Based Nutrition Program for Women and Children in Nepal Using Demographic and Health Survey

**DOI:** 10.3390/ijerph21060754

**Published:** 2024-06-09

**Authors:** Gauri Joshi, Masaru Ichihashi, Chalise Binaya

**Affiliations:** 1Graduate School of Humanities and Social Science, Hiroshima University, Higashihiroshima 739-8529, Japan; joshigauri22185@gmail.com (G.J.);; 2National Statistics Office, Kathmandu 44600, Nepal; 3The IDEC Institute, Hiroshima University, Higashihiroshima 739-8529, Japan; 4Graduate School of Innovation and Practice for Smart Society, Hiroshima University, Higashihiroshima 739-8529, Japan; 5Network for Education and Research on Peace and Sustainability (NERPS), Hiroshima University, Higashihiroshima 739-8529, Japan

**Keywords:** undernutrition, difference-in-differences method, weight-for-age z-score, weight-for-height z-score, height-for-age z-score

## Abstract

Undernutrition is a particularly acute problem in middle- and low-income countries. The “Suaahara” program is a 5-year community-focused program in Nepal, aimed at improving the health and nutrition of pregnant and lactating women and their children under the age of 2 years. This research contributes to evidence on the impact of the “Suaahara” program in 41 treated districts compared to 34 control districts. Using the difference-in-differences method, we found that the weight-for-height z-score and body mass index z-score of children under the age of 2 in the treated districts significantly increased by 0.223 standard deviations (SDs) and 0.236 SDs, respectively, compared with the control districts 5 years before and after the program. The number of antenatal care visits (at least four visits) and safe deliveries significantly increased for pregnant women by 10.4% and 9.1%, respectively, in the treated districts compared with the control districts. The prevalence of fever in children under 2 years of age was significantly reduced by 6.2% in the treated districts. The results show the significance of a policy evaluation with transparent indicators on public health, which is necessary for policymakers so that they can propose evidence-based policy.

## 1. Introduction

Investments in nutrition during the first 1000 days from pregnancy to a child’s second birthday are very crucial for babies [1,2]. Undernutrition has always been a major health concern, particularly in low- and middle-income countries. Childhood undernutrition may severely hamper the human and economic development of a country. Undernutrition during pregnancy leads to low birth weight, under-five child mortality, and impaired physical and cognitive development in children [3]. Undernutrition increases the risk of morbidity and mortality and perpetuates the intergenerational transmission of poverty, leading to declines in both long-term and short-term income-earning potential and labor force productivity [4]. The nutritional status of children is important during the first 1000 days of their lives, from conception to up to 2 years of age. Interventions to improve maternal nutrition during this period can lower the risk of health issues in mothers and their children. Nutrition education and counseling generate a greater impact if conducted in conjunction with other nutrition-sensitive interventions, such as nutrition safety nets or economic growth improvement programs [5].

Infant and young child feeding practices in Nepal are inadequate, contributing to the high prevalence of undernutrition [6]. However, additional work is required to promote access to proper nutrition and improve nutritional care to reduce stunting, wasting, and being underweight for all children, adolescent girls, and women of reproductive age.

Pokhara Declaration I was the first National Nutrition Strategy of Nepal, introduced in 1978, followed by the Second Nutrition Strategy (known as Pokhara Declaration II), introduced in 1986 [7]. The Joint Nutrition Support Program was a multi-sectoral nutrition program implemented in 1989–1992 that was ineffective owing to the low participation rate of all sectors. Later, in 2011, the National Planning Commission (NPC) developed the Nutrition Assessment and Gap Analysis (NAGA), which reported that undernutrition in Nepal was caused by poor food availability, lack of access and affordability, inadequate feeding and childcare practices, and poor food quality, leading to reduced nutrient density in food. The Multi-Sector Nutrition Plan (MSNP) 2013–2017 was implemented to improve children’s nutrition. The seven prioritized sectors in the MSNP 2013–2017 were health, agriculture, education, local development, water, sanitation and hygiene, and women and child welfare. Nepal is part of the Global Scaling Up Nutrition (SUN) Movement, which is dedicated to strengthening nutrition worldwide and has promised to end malnutrition in the country. To continue the successful efforts by safeguarding the achievements gained from the movement, Nepal formulated the MSNP II from 2018 to 2022, in which both nutrition-specific and nutrition-sensitive interventions were incorporated to address malnutrition by implementing methods that uniquely tackle the issue of undernutrition and benefit all segments of Nepalese society [8].

The present study serves as an indication for the country’s government and other stakeholders to consider continuing this type of nutritional intervention if improvements in health outcomes are achieved. This study assessed the impact of the “Suaahara” program on the health status of children under 2 years of age using the causal inference difference-in-differences (DID) method.

“Suaahara” is a Nepalese word that refers to good nutrition. The “Suaahara” program was a 5-year multifaceted community-based nutrition program, implemented in 41 districts in Nepal from August 2011 to August 2016 [9,10] (Figure 1). The program focused on improving the health and nutritional status of pregnant and lactating women (PLW) and children under 2 years of age. The primary fields of interest of the “Suaahara” program were gender equity and social inclusion, social and behavioral change communication, and social mobilization and governance. Evidence-based interventions, including training to plant and maintain small, diversified gardens and supporting poultry breeding to provide animal-source protein, were implemented to enhance the quality of health, nutrition, family planning, water, and sanitation. This project was financed by the United States Agency for International Development (USAID) in consultation with the Ministry of Health and Population. The “Suaahara” program concentrated on the first 1000 days of a child’s life, from the beginning of conception up to 2 years of age. The following map depicts the districts in which the “Suaahara” program was implemented.

The program included regular visits and follow-ups by Female Community Health Volunteers to households with PLW and children under 2 years of age. The “Suaahara” program included interactive groups for mothers to promote awareness regarding better nutritional health and sanitation practices for mothers and children. Effective counseling on healthy timing and spacing of pregnancies was provided. Communication campaigns, such as the “Mother Knows the Best” radio program, were broadcast to spread awareness and encourage behavioral changes in the health and nutrition practices of mothers in the first 1000 days. To enhance household food security and promote diversity in food cultivated and consumed at home, the program included training on creating a small, diversified garden. The program provided support for backyard poultry to obtain animal-based proteins and increase household income. To lower the incidence of infections, training and support were provided for the safe disposal of solid waste, frequent hand washing, and the construction of toilets.

The rest of this study is organized as follows:

After the Introduction section, the literature review is presented in Section 2, the data and methodology in Section 3, and the results in Section 4, followed by a robustness check in Section 5. Section 6 presents the discussion, and the conclusion is presented in Section 7.

## 2. Literature Review

Raza et al. [9] conducted a study in Bangladesh, where they implemented a multidimensional training program on entrepreneurship, health, nutrition, and social awareness, including the transfer of income-generating assets, in the Targeting Ultra-Poor (TUP) program. They found that these interventions resulted in improved child nutrition outcomes in weight-for-height z-score (WHZ) in participating households [11]. A study conducted in rural Guatemala reported that home garden interventions, including supplementation with garden material, monthly agricultural home visits, and agriculture classes when combined with nutrition-specific interventions, such as food supplementation with micronutrient powder, nutrition home visits, and group nutrition classes, improved children’s height/length-for-age z-scores and increased home crop production [12].

Osei et al. [13] assessed the Enhanced Homestead Food Production Program (EHFP) in Nepal, which provides gardening, animal husbandry, and nutritional education to beneficiary families, along with micronutrient powder (MNP) supplementation; they observed a significant reduction in anemia among children in the EHFP + MNP and EHFP groups compared with those in the control group. Olney, et al. [14] conducted a cluster-randomized controlled trial in Burkina Faso and reported that a health behavior changes communication program with an integrated agriculture program had statistically significant impacts on wasting, hemoglobin levels, anemia levels, and diarrhea in children aged 3–12.9 months old. In addition, they found statistically significant positive impacts on the participation of women in agriculture with agricultural production, women’s health, and nutrition-related knowledge and practices in treated villages compared with control villages. Iannotti et al. [15] conducted a randomized control trial in Ecuador and found that the consumption of eggs significantly increased the length-for-age z-score (HAZ) by 0.63 and the weight-for-age z-score (WAZ) by 0.61 in children aged 6–9 months. Awuuh et al. [16] reported a significant improvement in nutritional status among undernourished children aged 6–24 months in Ghana after a nutritional education intervention was provided to mothers. Improvements were noted in underweight, wasting, mid-upper arm circumference (MUAC), and hemoglobin levels among children following the nutrition education intervention. MUAC is commonly used in children aged 6–59 months as well as pregnant women to identify acute malnutrition. On a straight left arm, the MUAC is measured halfway between the tip of the shoulder and the tip of the elbow. Sisay and Tesfaye [17] stated that providing nutritional education and counseling to pregnant women improved maternal malnutrition and the rate of low birth weight in Ethiopia. Similarly, Alzua et al. [18] reported that a community-led total sanitation (CLTS) program exerted a significant positive impact on growth outcomes for children under the age of 5 years in Mali.

Previous studies have focused on conditional nutritional programs for specific communities, such as poor or ultra-poor households. However, the “Suaahara” program in Nepal focused on all PLW and children under 2 years of age in the treated districts. In addition, previous studies in Nepal included process evaluations using a simple quantitative approach. This study was based on impact evaluations using a DID econometric model.

## 3. Data and Methodology

This study used repeated cross-sectional data obtained from the Demographic and Health Survey (DHS) in 2001, 2006, 2011, and 2016 [19]. The DHS is a nationally representative household survey performed every 5 years that collects information on population-, health-, and nutrition-related monitoring and impact evaluation indicators.

The unit of analysis in this study was at the district level. Child growth statuses, such as WHZ, WAZ, body mass index z-score (BMIZ), and HAZ, were the primary outcome variables. The effects of continuous z-scores should be interpreted in terms of the standard deviations (SDs) from the median of the World Health Organization (WHO) international reference group. We analyzed the secondary outcomes and nutrition status of PLW, including WHZ, BMIZ, and Rohrer’s index (ROI). ROI measures the weight of the body per cubic unit of volume, assuming that the body is a three-dimensional cube, whereas the body mass index (BMI) measures the weight of the body per square unit of area, assuming that the body is a two-dimensional square sheet. Other secondary outcomes included antenatal care (ANC) visits, safe delivery, water and sanitation, and infections in children. The covariates in this study were household characteristics, as well as maternal and child characteristics. Household characteristics included family size, wealth index score, type of residence (urban/rural), and household head’s gender, religion, and caste. Maternal characteristics included age, years of schooling, smoking habits, and employment status. Child characteristics included gender, age, and size at birth.

The DID method was used to estimate the impact of the “Suaahara” program on the health status of children under 2 years of age and PLW. It measures the extent of changes from the pre-treatment period of an intervention to the post-treatment period in the treated group compared with that in the control group [20]. As there was no specific criterion for selecting the treated districts, the assumption was made that the program’s implementation could be likely treated as random. In order to confirm this, the parallel trend check is employed in this study. The quantitative data were divided into two periods: a pre-treatment period (DHS 2001, DHS 2006, DHS 2011, [19]) and a post-treatment period (DHS 2016, [19]), with 41 treated districts and 34 control districts included in the analysis.

The estimating equation for the DID method is:$$Y_{\mathrm{ijt}}=\beta_{0}+\beta_{1}Treatment_{j}+\beta_{2}\;\left(After_{t}\right)+\beta_{3}\left(Treatment_{j}*After_{t}\right)+\beta_{4}X_{ijt}+BY_{t}+D_{J}+u_{ijt,}$$
where $Y_{ijt}$ is the outcome variable for individual *i* (children under 2 years of age and PLW) in district *j* at time *t*. $X_{ijt}$ is the time-variant and -invariant covariate for individual *i* in district *j* at time *t*$D_{J}$ is the district fixed effect.$BY_{t}$ is the child’s birth year fixed effect.

The coefficients of the interaction term $(\beta_{3}$) show the impact of the “Suaahara” program in the treated districts compared to the control districts.

We checked the parallel trend assumption, which implied that if no treatment had occurred, the difference between the treated and control groups would have been the same in the post-treatment and pre-treatment periods. Since parallel trends were inherently unobservable, to test this hypothesis, we examined whether the treated and control groups had similar trajectories for the dependent variable before treatment (pre-treatment period). Consequently, we used a pre-treatment analysis.

The equation for measuring the pre-trend analysis is:$$Y_{ijt}=\beta_{0}+\overset{4}{\underset{i=1}{\sum }}\beta_{mi}Year_{t}+\overset{4}{\underset{i=1}{\sum }}\beta_{ni}Year_{t}*Treatment_{j}+\beta_{4}X_{ijt}+BY_{t}+D_{J}+u_{ijt,}$$
where the coefficient $\beta_{ni}$ indicates the trend of output (WHZ, WAZ, HAZ, and BMIZ) in the treated and control districts for the pre-treatment periods (2006 and 2011) compared to 2001.

A robustness check for this study was performed in two ways:A placebo test used a fake treatment period, in which two fake treatment periods (after 2006 and 2001) in the pre-treatment period were used, and the impact of this fake treatment period was evaluated.A placebo test was conducted using a Monte Carlo simulation, in which we randomly allocated individuals in the control group to the fake treatment (placebo) and control groups. We performed 1000 iterations of random allocation and performed a regression analysis using the same DID specifications with a randomly generated placebo (fake treatment) group.

Table 1 displays the data as descriptive statistics, including the mean difference between the treated and control districts before and after the intervention (columns 5 and 10, respectively).

## 4. Results

Table 2 is a summary table, which shows the average difference between the treatment group and control group in each indicator. Also, Table 3 displays the trends of the main outcome variables of children under 2 years of age and PLW, respectively, showing coefficients in each period, such as 2006, 2011, and 2016. HAZ, WAZ, WHZ, BMIZ, and ROI are checked here. According to the F-test, since the hypothesis that coefficients of β_n2_ in 2006 and β_n3_ in 2011 are the same is not rejected, the parallel trends assumption is acceptable. Regarding trends of the level in these indicators before and after the program, the parallel trend can clearly be seen (Appendix A).

A notable finding in Table 4 shows that the “Suaahara” program significantly increased children’s WHZ and BMIZ. This is because the program focused on improving health- and nutrition-related knowledge and practices, which ultimately improved the indicators of acute malnutrition (WHZ) in the target districts. The findings from the DID estimation indicated that the WHZ of children under 2 years of age increased by 0.223 SDs in the treated districts compared to the control districts, at a 5% level of significance. Similarly, the BMIZ of children increased by 0.236 SDs in the treated districts compared to the control districts, at a 5% level of significance. These findings suggested that the positive impact of the program was insignificant for HAZ and WAZ in children under 2 years of age. Moreover, we did not find significant results for the nutrition status of PLW (WHZ, BMIZ, and ROI), as shown in Table 3.

The access of pregnant women to health institutions increased in the treated districts. We observed that ANC visits (at least four times) to a health institution for pregnant mothers were statistically significant (1% level of significance) and increased by 10.4% in the treated districts compared to the control districts. In addition, safe deliveries increased by 9.1%, and the result was statistically significant at the 1% level. In addition, the prevalence of fever in children younger than 2 years of age during the 2 weeks prior to the DHS was reduced by 6.2% in the treated districts compared to the control districts, and the results were statistically significant (at the 5% level of significance). However, the result was insignificant for the prevalence of diarrhea in children under 2 years of age (Table 5).

The program resulted in increased water and sanitation facilities in the treated districts. The proportion of children with improved and basic water sources in the household increased, with statistically significant results of 10.4% and 10.2%, respectively, in the treated districts than in the control districts. Households with drinking water sources, such as household pipe connections, public taps/standpipes, tube wells/boreholes, protected wells, protected springs, rainwater, tanker trucks, and carts with small tanks or bottled water, were considered as households with improved water sources. Households with improved water sources, which had either a water source on the premises or a round-trip water collection time of 30 min or less, were categorized as basic water source households [21]. Similarly, the use of improved sanitation increased by 19.6%, and the result was significant; however, it was insignificant for the use of basic sanitation facilities (Table 6). Households with a toilet that can flush to a piped sewer system, or a septic tank, a pit toilet, a pit toilet with a ventilated improved pit (VIP), a pit toilet with a slab, or a composting toilet were classified as households with improved sanitation. Households with improved sanitation that was not shared with other households were classified as households with basic sanitation services [21]. 

## 5. Robustness Check

In the placebo test for the fake treatment periods, the first fake treatment was provided after 2006 (the fake post-treatment period was 2011) and the other fake treatment was administered after 2001 (the fake post-treatment periods were 2006 and 2011). We conducted the DID for two fake treatments, and the results demonstrated that the placebo effects in rows 1 and 3 were insignificant for HAZ, WAZ, WHZ, and BMI of the children, indicating no DID effect. Therefore, the outcome variables in the treated group did not change during the pre-treatment period (Appendix B).

The result reported in the given table (Appendix C) presents the mean values obtained from the 1000 iterations, which provided insignificant results. Hence, the control districts did not change and were not impacted by the “Suaahara” program.

From two placebo tests, we concluded that the change in the treated group occurred due to the impact of the “Suaahara” program.

Figure 2 displays the distribution of the placebo effect, and it is evident that the frequency of the placebo effect with a value of zero or near zero is high.

## 6. Discussion

This study is a pioneering work, showing the impact of nutritional interventions on the acute and chronic nutritional status of children and PLW. WAZ is a chronic nutritional indicator. It measures chronic malnutrition, which is a long-term developmental concern and requires a substantial period of time for improvement. WHZ is an acute nutritional indicator that measures acute malnutrition when an individual suffers from current, severe nutritional restrictions, a recent bout of illness, inappropriate child practices, or a combination of these factors. Since the program is relatively short, lasting for only 5 years, and also children under 2 years usually drastically grow, the reason for different results between WAZ and WHZ is not clear to us so far. This study only provided a basis for monitoring and evaluating currently implemented interventions, which is a limitation.

Investments in nutrition intervention-related programs for PLW and children that increase nutritional health are necessary, which will ultimately improve the health of productive populations. This requires complementary partnerships with the national government and other stakeholder agencies to work as effectively as possible to strengthen agricultural production and health systems and provide water, sanitation, and hygiene facilities.

Therefore, evidence-based nutritional interventions that reduce maternal and child mortality rates in low- and middle-income countries should be included in key policies to improve the quality of life of children and PLW.

The study has several limitations that should be acknowledged. The presence of other nutrition programs that were not considered in this study could have impacted the nutritional health of children and PLW. The unavailability of certain household characteristics and statistical data that affect the health of children and PLW is another limitation. Although no criteria exist for the selection of districts for the “Suaahara” program, in a low-income country, such as Nepal, factors such as political influence and the socioeconomic structure of the districts may have introduced selection bias. Finally, additional research is required to confirm the validity of these findings by examining the effects of community-based nutrition programs on the health of children and PLW.

Further studies are required to evaluate the impact of community-based nutritional interventions on women’s empowerment. In addition, the effects of this nutritional intervention on children’s cognitive development should be analyzed.

## 7. Conclusions

Participating in nutritional practices in the “Suaahara” program, including health counseling, crop production from home-based gardening, improved clean water and sanitation practices, and raising backyard poultry, increases the nutritional health of PLW and children under 2 years of age.

Using the DID method, we mainly found the following points: The WHZ of children under 2 years of age significantly improved by 0.223 SDs in the “Suaahara”-treated districts than in the control districts. Similarly, the intervention significantly increased the BMIZ of children under 2 years of age by 0.236 SDs. The number of pregnant women with at least four ANC visits at health institutions increased in the “Suaahara”-treated districts by 10.4%, and this result was statistically significant at the 5% level of significance. The number of women with safe deliveries increased by 9.1% in the treated districts compared to in the control districts. The “Suaahara” program also had a positive impact on the health of young children, as the prevalence of fever decreased by 6.2% in children under 2 years in the treated districts compared to in the control districts. Moreover, water and sanitation services increased in the treated districts. The number of households with children under 2 years of age with improved water sources, basic water sources, and improved sanitation increased by 10.4%, 10.2%, and 19.6%, respectively, in the treated districts compared with that in the control districts.

Considering all of this, we can summarize that this program could improve the health condition of both infants and their mothers significantly in terms of some factors, like ANC visits, safe deliveries, the prevalence of fever in children, and so on, in Nepal. It shows that this kind of healthcare policy in low- and middle-income countries is crucial for human development. Nutrition education, awareness, and income-generating activity in this program improve nutritional practices for women and children. The most valuable resource for any nation is its children, whose physical, mental, and emotional growth will be vital for them as they grow up and become citizens. The health of children is greatly influenced by the mothers’ knowledge of child nutrition [22].

At the least, if they introduce the healthcare policy, the evaluation of the policy on how much it affects the objectives is necessary for policymakers. Clarifying whether public policies or programs are effective or not is very important for any country facing budget constraints now and in the future. Also, this process is essential for policymakers so that they can propose evidence-based policy.

## Figures and Tables

**Figure 1 ijerph-21-00754-f001:**
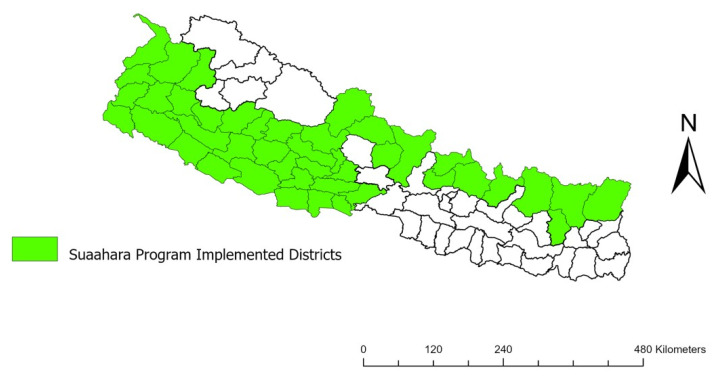
“Suaahara” program implemented districts (source: Survey Department, Ministry of Land Management, Cooperatives and Poverty Alleviation, Nepal).

**Figure 2 ijerph-21-00754-f002:**
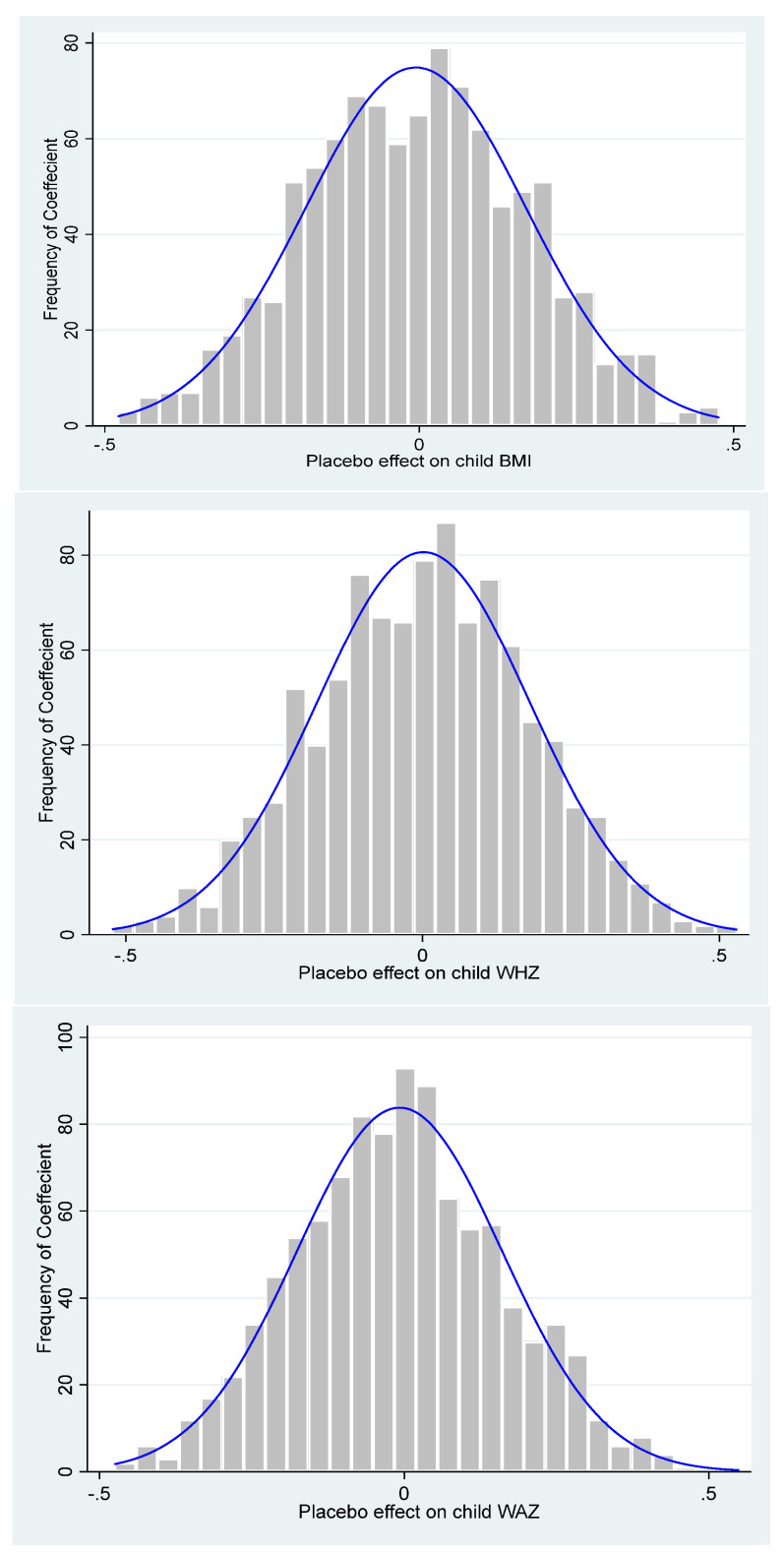
Distribution of placebo effect.

**Table 1 ijerph-21-00754-t001:** Descriptive statistics.

Variables	Before					After				
	Treatment		Control			Treatment		Control		
	(1)	(2)	(3)	(4)	(5)	(6)	(7)	(8)	(9)	(10)
	Obs.	Mean	Obs.	Mean	Diff (2)–(4)	Obs.	Mean	Obs.	Mean	Diff (7)–(9)
Family size	3586	6.857	2509	6.577	0.281 ***	930	6.158	807	6.384	−0.226
		−0.057		−0.058	−0.081		−0.095		−0.102	−0.139
Mother Education Year	3586	2.604	2509	2.709	−0.106	930	5.952	807	4.918	1.033 ***
		−0.062		−0.077	−0.098		−0.134		−0.152	−0.203
Father Education Year	3567	5.343	2469	5.021	0.322 ***	929	7.115	804	6.693	0.422
		−0.066		−0.085	−0.108		−0.115		−0.134	−0.177
Mother Employment	3586	0.814	2509	0.579	0.234 ***	930	0.546	807	0.367	0.179 ***
		−0.006		−0.009	−0.012		−0.016		−0.017	−0.024
Female Child	3586	0.493	2509	0.498	−0.005	930	0.468	807	0.437	0.031
		−0.008		−0.009	−0.013		−0.016		−0.017	−0.024
Age of child (in months)	3586	11.798	2509	11.675	0.123	930	12.135	807	12.114	0.021
		−0.113		−0.134	−0.176		−0.222		−0.238	−0.326
Poverty	3586	0.505	2509	0.354	0.150 ***	930	0.573	807	0.365	0.207 ***
		−0.008		−0.009	−0.013		−0.016		−0.017	−0.023
Low birth size	3586	0.255	2506	0.165	0.091 ***	927	0.202	807	0.136	0.065 ***
		−0.007		−0.007	−0.011		−0.013		−0.012	−0.018
Religion (Hindu)	3586	0.901	2509	0.782	0.119 ***	930	0.889	807	0.818	0.071 ***
		−0.005		−0.008	−0.009		−0.011		−0.014	−0.017
Rural residence	3586	0.135	2509	0.194	−0.058 ***	930	0.561	807	0.571	−0.009
		−0.006		−0.008	−0.009		−0.016		−0.017	−0.024
Disadvantaged Caste	3586	0.536	2509	0.724	−0.187 ***	930	0.577	807	0.747	−0.169 ***
		−0.008		−0.009	−0.012		−0.016		−0.015	−0.022
Female Household head	3586	0.187	2509	0.145	0.0421 ***	930	0.316	807	0.249	0.067 ***
		−0.006		−0.007	−0.009		−0.015		−0.015	−0.022
Nonsmoking Mother	3586	0.807	2509	0.811	−0.004	930	0.936	807	0.964	−0.027 ***
		−0.006		−0.007	−0.01		−0.008		−0.006	−0.011
Mother Age (years)	3586	25.868	2509	25.956	−0.087	930	24.909	807	25.037	−0.127
		−0.098		−0.118	−0.154		−0.174		−0.191	−0.257

Robust clustered standard error in parentheses. *** *p* < 0.01.

**Table 2 ijerph-21-00754-t002:** Summary table of difference between treatment and control in each indicator.

Outcome	2001					2006					2011					2016				
	Treatment		Control			Treatment		Control			Treatment		Control			Treatment		Control		
	Observation	Mean	Observation	Mean	Difference (2)–(4)	Observation	Mean	Observation	Mean	Difference (7)–(9)	Observation	Mean	Observation	Mean	Difference (12)–(14)	Observation	Mean	Observation	Mean	Difference (17)–(19)
	(1)	(2)	(3)	(4)	(5)	(6)	(7)	(8)	(9)	(10)	(11)	(12)	(13)	(14)	(15)	(16)	(17)	(18)	(19)	(20)
WHZ	1449	−0.882	1156	−1.024	0.141 ***	1249	−1.041	850	−1.020	−0.021	532	−0.841	398	−0.764	−0.077	551	−0.565	447	−0.837	−0.271 ***
		(0.031)		(0.037)	(0.048)		(0.034)		(0.040)	(0.053)		(0.053)		(0.069)	(0.087)		(0.053)		(0.061)	(0.081)
BMIZ	1450	−0.791	1158	−0.933	0.142 ***	1254	−0.926	851	−0.928	0.002	532	−0.785	401	−0.725	−0.060	554	−0.458	451	−0.783	0.325 ***
		(0.030)		(0.038)	(0.048)		(0.034)		(0.040)	(0.053)		(0.054)		(0.068)	(0.087)		(0.054)		(0.062)	(0.083)
WAZ	1457	−1.622	1173	−1.687	0.066	1266	−1.612	860	−1.410	−0.201 ***	533	−1.256	403	−1.234	−0.022	554	−0.878	450	−1.022	0.144 *
		(0.031)		(0.036)	(0.048)		(0.033)		(0.043)	(0.055)		(0.055)		(0.059)	(0.081)		(0.053)		(0.059)	(0.079)
HAZ	1448	−1.729	1152	−1.640	−0.089	1258	−1.564	853	−1.294	−0.270 ***	531	−1.180	397	−1.124	−0.056	551	−0.909	449	−0.779	−0.130
		(0.037)			(0.057)		(0.039)		(0.052)	(0.065)		(0.066)		(0.077)	(0.101)		(0.068)		(0.069)	(0.097)
Rohrer’s index	1553	13.472	1248	13.344	0.127 **	1351	13.449	909	13.434	0.015	554	13.670	431	14.038	−0.368 ***	572	14.147	464	13.943	0.204
		(0.039)		(0.048)	(0.062)		(0.045)		(0.061)	(0.076)		(0.082)		(0.095)	(0.125)		(0.097)		(0.104)	(0.071)
Atleast4ANCvisit	1498	0.449	1208	0.532	−0.083 ***	1299	0.717	880	0.791	−0.073 ***	1217	0.582	820	0.528	0.053 ***	1065	0.750	942	0.676	0.074 ***
		(0.013)		(0.014)	(0.019)		(0.012)		(0.014)	(0.018)		(0.014)		(0.017)	(0.022)		(0.013)		(0.015)	(0.021)
Diarrhea	1462	0.265	1184	0.291	−0.027	1274	0.197	874	0.174	0.023	1204	0.214	817	0.207	0.007	1070	0.093	940	0.096	0.013
		(0.011)		(0.013)	(0.017)		(0.011)		(0.013)	(0.017)		(0.012)		(0.014)	(0.018)		(0.008)		(0.009)	(0.805)
Feverinlasttwoweeks	1462	0.371	1184	0.396	−0.025	1274	0.244	874	0.207	0.037 **	1204	0.248	817	0.223	0.026	1070	0.209	940	0.250	−0.041 **
		(0.013)		(0.014)	(0.019)		(0.012)		(0.013)	(0.018)		(0.012)		(0.014)	(0.019)		(0.012)		(0.014)	(0.018)

The number of parentheses is the standard error. *** *p* ≤ 0.01, ** *p* ≤ 0.05, * *p* ≤ 0.1.

**Table 3 ijerph-21-00754-t003:** Pre-trend analysis.

Variables	Children under 2 Years of Age				PLW		
	(1)	(2)	(3)	(4)	(5)	(6)	(7)
	HAZ	WAZ	WHZ	BMIZ	WHZ	BMIZ	ROI
β_m2_ (Year 2006)	0.397	−0.021	−0.719 *	−0.519	0.305	−0.047	0.022
	−0.462	−0.336	−0.375	−0.366	−3.305	−0.587	−0.405
β_m3_ (Year 2011)	1.083	−0.305	−1.955 **	−1.285	3.514	0.389	0.379
	−0.861	−0.588	−0.919	−0.794	−6.119	−1.097	−0.754
β_m4_ (Year 2016)	1.033	0.792	−0.03	0.348	6.737	0.846	0.772
	−0.878	−0.723	−0.773	−0.788	−9.523	−1.705	−1.173
β_n2_ (Year 2006 × Treatment)	−0.147	−0.200 **	−0.101	−0.078	−1.23	−0.295	−0.172
	−0.112	−0.089	−0.085	−0.086	−1.065	−0.189	−0.129
β_n3_ (Year 2011 × Treatment)	0.065	−0.075	−0.236 *	−0.217 *	−2.375 *	−0.457 **	−0.318 **
	−0.139	−0.113	−0.123	−0.127	−1.362	−0.223	−0.16
β_n4_ (Year 2016 × Treatment)	−0.039	0.064	0.148	0.172	−0.855	−0.184	−0.113
	−0.132	−0.104	−0.112	−0.114	−1.683	−0.296	−0.205
Observations	5851	5903	5843	5859	5895	5927	5927
R-squared	0.33	0.311	0.17	0.161	0.179	0.193	0.168
F test Ho: β_m2_ = β_m3_	0.802	0.245	2.75	1.455	2.172	1.838	2.009
F test Ho: β_n2_ = β_n3_	1.443	2.541 *	2.01	1.531	1.631	2.364 *	2.108

Robust clustered standard error in parentheses. ** *p* < 0.05, * *p* < 0.1.

**Table 4 ijerph-21-00754-t004:** Impact of the “Suaahara” program on the growth status and the nutrition status.

Variables	Children under 2 Years of Age				PLW		
	(1)	(2)	(3)	(4)	(5)	(6)	(7)
	HAZ	WAZ	WHZ	BMIZ	WHZ	BMIZ	ROI
Treatment Effect	0.009	0.153	0.223 **	0.236 **	0.154	0.033	0.023
	−0.119	−0.094	−0.104	−0.106	−1.572	−0.278	−0.191
Observations	5851	5903	5843	5859	5895	5927	5927
R-squared	0.328	0.307	0.167	0.158	0.175	0.188	0.164

Robust clustered standard error in parentheses. ** *p* < 0.05.

**Table 5 ijerph-21-00754-t005:** Impact of the “Suaahara” program on the infection of children under 2 years and usage of ANC and safe delivery by PLW.

Variables	Children under 2 Years of Age		PLW	
	(1)	(2)	(3)	(4)
	Diarrhea	Fever	ANC	Safe Delivery
Treatment Effect	−0.015	−0.062 **	0.104 ***	0.091 ***
	−0.021	−0.028	−0.034	−0.032
Observations	7763	7763	7615	7725
R-squared	0.08	0.081	0.253	0.401

Robust clustered standard error in parentheses. *** *p* < 0.01, ** *p* < 0.05.

**Table 6 ijerph-21-00754-t006:** Impact of the “Suaahara” program on water and sanitation facilities.

Variables	(1)	(2)	(3)	(4)
	Improved Water Source	Basic Water Source	Improved Sanitation	Basic Sanitation
Treatment Effect	0.104 ***	0.102 ***	0.196 ***	0.037
	−0.025	−0.026	−0.035	−0.039
Observations	7768	7076	7768	3771
R-Squared	0.263	0.264	0.529	0.281

Robust clustered standard error in parentheses. *** *p* < 0.01.

## Data Availability

In this study, there is no generated data.

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
