# Peer review of "Assessment of a Community-Based Nutrition Program for Women and Children in Nepal Using Demographic and Health Survey"

_ijerph, 2024, doi:10.3390/ijerph21060754_

Round 1
Reviewer 1 Report
Comments and Suggestions for Authors
It is an interesting study. Kindly respond to my comments below:
I suggest instead of adding the number of references at the beginning of the sentence, add the surname of the author followed by et al. or co-workers and write a complete, meaningful sentence. This applies to L 125, 131, 134, etc).
Line 211, 226, 239, and 251: text under the Table. There are stars which weren't there, but still, the text under the table referred to them
On what basis were the centres selected for inclusion in the program?
What factors led to the success of the program? What are the lessons learnt?
The Conclusion section repeats the result section rather than an overall look at the study findings without repeating values. It also includes the study recommendation.
What are the recommendations of the study?
No referral to Appendix A and B was made in the text.
How was the education performed during the intervention period?
What was the characteristics of the educators or researchers who implemented the program?
What message was given, in which language, how many times during the program, and in which place?
Did the educators receive any training, and for how long?
From where was the ethical permission obtained?
Author Response
Referee 1:
- I suggest instead of adding the number of references at the beginning of the sentence, add the surname of the author followed by et al. or co-workers and write a complete, meaningful sentence. This applies to L 125, 131, 134, etc.).
Response:
Thank you for your pointing out. We’re not so familiar with the reference style of this journal. We added the surname of the authors in each reference number.
- Line 211, 226, 239, and 251: text under the Table. There are stars which weren't there, but still, the text under the table referred to them
Response:
Thank you for this comment, but I’m not so sure about the meaning of it. Each table has estimation results where there are some numbers with asterisks and others without. All tables have these kinds of numbers so we put the note about the general information about the asterisks.
- On what basis were the centres selected for inclusion in the program?
Response:
Authorities in Nepal who conducted this survey selected the district and individuals. In this research, we used secondary data from the survey so we followed their selection standard, which means that we didn’t set up any basis on our own.
- What factors led to the success of the program? What are the lessons learnt?
Response:
Thank you for your question.
The Suaahara program had not been evaluated in terms of statistical figures on what factors contributed to the program. In this paper, we tried to clarify the impact of the program on infants and their mothers with numerical data, which is an academic paper for the first time.
- The Conclusion section repeats the result section rather than an overall look at the study findings without repeating values. It also includes the study recommendation.
Response:
Thank you for your comments. we added a brief policy implication and recommendation as well as the main findings in the conclusion section, mentioning
“Above these, we can summarize that this program could improve the health and nutrition condition of both infants and their mothers significantly around 6 % to 10% of health indicators in Nepal. It shows that this kind of health care policy in low- and middle-income countries is crucial for human resources development. At least, if they introduce the healthcare policy, the evaluation of the policy on how much it affected objectives is necessary for policymakers. Clarifying whether public policy or program is effective or not is very important for any country facing a budget constraint now and in the future.”
- What are the recommendations of the study?
Response:
This study’s recommendation is to show the significance of policy evaluation with empirical evidence. Clarifying whether public policy or program is effective or not is very important for any country facing a budget constraint.
We added a brief mention into the conclusion section as told above.
- No referral to Appendix A and B was made in the text.
Response:
We already mentioned Appendix A and B in the text (lines 263-264), but if the referee means we should put information about references in Appendix A&B, we have not used particular references in this part.
- How was the education performed during the intervention period?
Response:
Thank you for your comments.
But we don’t have any information about the education during the program period.
This point might be a limitation of this research.
- What was the characteristics of the educators or researchers who implemented the program?
Response:
Thank you for your comments. Similarly, we don’t have any information about the characteristics of the educators and researchers on this program.
- What message was given, in which language, how many times during the program, and in which place?
Response:
Thank you for your inquiry.
Messages regarding nutrition education for PLW and children are given which are in the Nepali language and also in the local language. The messages are given through a radio program, so we don’t have any idea of frequency of it. FCHV and the mother's group also helped in giving the message.
- Did the educators receive any training, and for how long?
Response:
Thank you for your comments. Educators received training but we don’t have any information regarding the period of training.
- From where was the ethical permission obtained?
Response:
This Suaahara program was implemented by the central and local governments so they didn’t seem to need to obtain such ethical permission.

Reviewer 2 Report
Comments and Suggestions for Authors
This study uses and cross-sectional difference in difference approach to estimate the impacts of the Suaahara on child and maternal nutritional status as well as child morbidity, ANC care, and water and sanitation-related outcomes. Overall, I appreciate the study but feel that there are a number of ways in which the study (and presentation and interpretation of results) can be improved. My comments are listed below.
Title: The title doesn’t seem to reflect the study, which assesses the impact of a nutrition program on the nutritional status of children and PLW (in additional to other outcomes). You might consider revising to better reflect what you actually assessed.
Line 21: The research doesn’t contribute to the impact. You might consider rephrasing to something like “This research contributes to the evidence base on the impact of…”
Lines 22-23: Since the acronyms DID, WHZ, and BMIZ are not used again in the abstract, you can save words by dropping the acronyms. Same with ANC later in the abstract.
Lines 22-24: You need to specify for which group this impact was observed (children or PLW).
Line 24: If you have room, please specify the period over which the increase was observed.
Abstract: I think the abstract would be stronger if you can rework it a bit to (a) briefly describe what the Suaahara program entailed (i.e., what interventions were implemented to improve the health and nutrition of PLW and young children) and (b) at the end, describe the policy relevance of your findings.
Line 37: Please add references to this claim, and describe what, specifically, you mean by investments in nutrition being highly “fruitful.” Fruitful in terms of what, specifically?
Line 39: Please consider replacing “developing countries” with “low- and middle-income countries” (here and elsewhere).
Line 83: “Economic growth” is not really an intervention, rather it’s the goal of an intervention/set of interventions and/or policies. Can you be more specific here about what the program included in an effort to spur economic growth?
Line 94: Was it and/or or did eligible households have to have both a PLW and child under 2?
Lines 110-148: I think this literature review section might be more effective if integrated into the discussion section. That is, you can use parts of the text from this literature review section to place your results in the context of existing evidence and to identify how your results add to the existing evidence (as you’ve noted in lines 143-148).
Line 170: Why not also asses the impacts among older children (e.g., children 5-59 months)? Children under age 5 in 2016 would have been exposed to the treatment during their first 1000 days, right?
Lines 173-174: Though the “before” statistics in table 1 would seem to indicate that the program’s implementation was not at all random, as the treatment and control groups vary significantly across many different dimensions (poverty, father’s education, urban/rural residence, low birth size, sex of head of household, etc). How do you square your assumption of random assignment with these statistics?
Line 175: Important to specify here that the 2011 DHS data collection was completed BEFORE the implementation of the program in Aug of 2011 (I had to look this up to confirm).
Results. In the results section, please also simply present your outcomes for children and PLW by DHS round and treatment group (i.e., mean HAZ, WAZ, WHZ, ANC visits, fever, diarrhea, water and sanitation outcomes, etc). This will help readers understand the baseline situation and to interpret the treatment effects.
Line 236: For adults, I don’t think we’d refer to these indicators as “growth status” but rather nutritional status (as these women are done growing).
Lines 240-243: As you saw improvements in ANC visits, it seems like there may have been positive impacts on birth outcomes. Any reason you didn’t assess, e.g., birth weight or other birth outcomes?
Robustness checks. As you are presenting impact of the program on a range of other outcomes, why did you not also apply the placebo tests for the other outcomes (maternal nutritional status, child infection, ANC visits, water and sanitation, etc)? Please either add these robustness checks or justify why you did not include them.
Lines 283-284: This explanation does not make sense to me. The program began in 2011, so all of the children under age 2 in the 2016 DHS survey were exposed to this intervention for their entire life up to the point of the survey, which I would surely not categorize as a short-term intervention.
A quick google search brought up this study that also assessed the impacts of the intervention using data collected during the intervention rather than DHS data (https://onlinelibrary.wiley.com/doi/full/10.1111/mcn.13630). It seems like it is also important to discuss your findings in the context of these findings, which differ (and any other findings related to the impacts of this program that have been published).
Lines 292-293: These are very sweeping claims. Please limit claims to those that can actually be supported by your findings. The only impact on morbidity was on the prevalence of fever, and you did not show any impacts on mortality.
Lines 296-398: Again, please limit claims to those actually evaluated by the study. You did not assess the impact of the intervention on healthcare costs, women’s SES, or women’s empowerment.
Line 299: Again, you did not assess mortality rates. This entire paragraph is not supported by your findings.
Lines 315-330: This section is not really conclusions but rather a line-by-line restatement of the findings. To me, an effective conclusions section should provide a concise summary of the key findings and, importantly, why those findings are important and how they can help inform policy.
Comments on the Quality of English LanguageThe quality English Language is adequate.
Author Response
Referee 2:
This study uses and cross-sectional difference in difference approach to estimate the impacts of the Suaahara on child and maternal nutritional status as well as child morbidity, ANC care, and water and sanitation-related outcomes. Overall, I appreciate the study but feel that there are a number of ways in which the study (and presentation and interpretation of results) can be improved. My comments are listed below.
Response:
Thank you very much for your valuable comments and suggestions. We’ve revised the draft basically according to the referee’s comments to clarify some ambiguous points.
- Title: The title doesn’t seem to reflect the study, which assesses the impact of a nutrition program on the nutritional status of children and PLW (in additional to other outcomes). You might consider revising to better reflect what you actually assessed.
Response:
Thank you for your suggestion. According to it, we revised the title:
“Assessment by Mass Health Indicators of a Community-Based Nutrition Program for Children in Nepal”
- Line 21: The research doesn’t contribute to the impact. You might consider rephrasing to something like “This research contributes to the evidence base on the impact of…”
Response:
Thank you for your suggestion. According to it, we rephrased that sentence into “This research contributes to the evidence based on the impact of the “Suaahara” program in 41 treated districts compared to 34 control districts”. (line 21)
- Lines 22-23: Since the acronyms DID, WHZ, and BMIZ are not used again in the abstract, you can save words by dropping the acronyms. Same with ANC later in the abstract.
Response:
Thank you for your comment.
We thought we’d better mention these acronyms in the abstract because those will be used in the body, but according to your comment, we dropped them from the abstract.
- Lines 22-24: You need to specify for which group this impact was observed (children or PLW).
Response:
Thank you for your comment.
We clarified this impact was observed in children under the age of 2.
- Line 24: If you have room, please specify the period over which the increase was observed.
Response:
Thank you for your comment.
According to the comment, we revised the sentence as
“Using the difference-in-differences method, we found that the weight-for-height z-score and body mass index z-score of children under the age of 2 in the treated districts significantly increased by 0.223 standard deviations (SDs) and 0.236 SDs, respectively, compared with the control districts during 5 years before the program in 2011 and after the program in 2016.” (lines 22-26)
- Abstract: I think the abstract would be stronger if you can rework it a bit to (a) briefly describe what the Suaahara program entailed (i.e., what interventions were implemented to improve the health and nutrition of PLW and young children) and (b) at the end, describe the policy relevance of your findings.
Response:
Thank you for your comment.
We revised the abstract to shorten the results and to add a brief policy implication as follows:
“The result shows the significance of policy evaluation with transparent indicators on public health which is necessary for policymakers to propose evidence-based policy making.” (lines 29-31)
- Line 37: Please add references to this claim, and describe what, specifically, you mean by investments in nutrition being highly “fruitful.” Fruitful in terms of what, specifically?
Response:
Thank you for your comment.
Line 37 : We added two references as :
Jean-Louis Arcland, Undernourishment and Economic Growth, 2001, Rome : FAO, 1-63, https://www.fao.org/4/X9280E/x9280e02.htm#b and
International Food Policy Research Institute(IFPRI) and United Nations System Standing Committee on Nutrition, Nutrition: A Foundation for Development, 2002, 1-52, Brief1-12EN.pdf (unscn.org)
https://www.unscn.org/layout/modules/resources/files/Brief1-12EN.pdf.
And we also revised the first sentence as follows:
“Investments in nutrition are highly fruitful because investment in nutrition during first 1000 days from pregnancy to a child's second birthday promotes health, reducing poverty and achieving long-term economic growth [1,2]”. (lines 36-37)
- Line 39: Please consider replacing “developing countries” with “low- and middle-income countries” (here and elsewhere).
Response:
Thank you for your comment.
We replaced “developing countries” with “low- and middle-income countries” or “a low-income country” (line 39, and elsewhere).
- Line 83: “Economic growth” is not really an intervention, rather it’s the goal of an intervention/set of interventions and/or policies. Can you be more specific here about what the program included in an effort to spur economic growth?
Response:
Thank you for your comment.
Yes, the referee’s point is right. The goal of Suaahara is to promote economic growth. To spur economic growth, the program provides training to plant and maintain small, diversified gardens and supports poultry breeding to provide animal-source protein and increase incomes.
So, we revised the expression here (line 83), saying
“Evidence-based interventions including training to plant and maintain small, diversified gardens and supporting poultry breeding to provide animal-source protein were implemented to enhance the quality of health, nutrition, family planning, water, and sanitation”.
- Line 94: Was it and/or or did eligible households have to have both a PLW and child under 2?
Response:
Yes, this program targets households with PLW and children under 2 years of age.
- Lines 110-148: I think this literature review section might be more effective if integrated into the discussion section. That is, you can use parts of the text from this literature review section to place your results in the context of existing evidence and to identify how your results add to the existing evidence (as you’ve noted in lines 143-148).
Response:
Thank you for your comment.
We think holding the Literature Review part and Discussion part would be good for a balance of this paper’s structure. So, there would not be a serious problem even if we still have this literature review part.
- Line 170: Why not also asses the impacts among older children (e.g., children 5-59 months)? Children under age 5 in 2016 would have been exposed to the treatment during their first 1000 days, right?
Response:
Thank you for your comment.
Yes, children under age 5 have also been exposed to the treatment, which started in August 2011, but this five-year is relatively long for infants to grow. Their growth might be affected by many factors, which are tougher to control factors in the estimation. So, this time, we focused on only shorter-period babies who might be sharply influenced by the treatment.
- Lines 173-174: Though the “before” statistics in table 1 would seem to indicate that the program’s implementation was not at all random, as the treatment and control groups vary significantly across many different dimensions (poverty, father’s education, urban/rural residence, low birth size, sex of head of household, etc.). How do you square your assumption of random assignment with these statistics?
Response:
Thank you for your comment.
To justify our assumption in this study, we used the pre-trend test as shown on the body. But as you pointed out, this sentence looks not so precise as an expression, so we revised this to express as
“the assumption was made to that the program’s implementation could be likely treated as random. In order to confirm this regard, the parallel trend check is employed in this study.”(lines 173-176)
- Line 175: Important to specify here that the 2011 DHS data collection was completed BEFORE the implementation of the program in Aug of 2011 (I had to look this up to confirm).
Response:
Thank you very much for confirming.
As per annual data, DHS 2011 is involved in the pre-treatment period. That’s why we mainly treated DHS 2016 as the treatment period.
- In the results section, please also simply present your outcomes for children and PLW by DHS round and treatment group (i.e., mean HAZ, WAZ, WHZ, ANC visits, fever, diarrhea, water and sanitation outcomes, etc). This will help readers understand the baseline situation and to interpret the treatment effects.
Response:
Thank you very much for your comments.
We revised the beginning of the results to make it much simpler as follows:
“Table 2 displays the trends of the main outcome variables of children under 2 years of age and PLW respectively showing coefficients in each period such as 2006, 2011, and 2016. Variables HAZ, WAZ, WHZ, BMIZ, and ROI are checked here. According to the F-test, since the hypothesis that coefficients of βn2 in 2006 and βn3 in 2011 are the same is not rejected, which means the parallel trends assumption is acceptable.” (lines 216-220)
- Line 236: For adults, I don’t think we’d refer to these indicators as “growth status” but rather nutritional status (as these women are done growing).
Response:
Thank you very much for your comments.
According to your suggestion, we have replaced “growth status” of PLW with “nutrition status”. (lines 159, 232 and 234)
- Lines 240-243: As you saw improvements in ANC visits, it seems like there may have been positive impacts on birth outcomes. Any reason you didn’t assess, e.g., birth weight or other birth outcomes?
Response:
Thank you very much for your comments.
Yes, of course, the program may have been positively impacting other birth outcomes. In this study, we employed limited variables like WHZ, BMIZ, ROI, ANC, and Safe Delivery, but we’d like to try other factors in another opportunity.
- Robustness checks. As you are presenting impact of the program on a range of other outcomes, why did you not also apply the placebo tests for the other outcomes (maternal nutritional status, child infection, ANC visits, water and sanitation, etc.)? Please either add these robustness checks or justify why you did not include them.
Response:
Thank you very much for your comments.
In this study, we mainly focused on variables like WHZ, BMIZ, ROI, ANC, and Safe Delivery, so we also limited some variables for the placebo tests to avoid the redundancy of the paper.
- Lines 283-284: This explanation does not make sense to me. The program began in 2011, so all of the children under age 2 in the 2016 DHS survey were exposed to this intervention for their entire life up to the point of the survey, which I would surely not categorize as a short-term intervention.
Response:
Thank you very much for your comments.
Yes, what you mentioned sounds reasonable. We revised this part to make a more mild interpretation as follows:
“Since the program is relatively short, lasting for only 5 years, and also children under 2 years usually drastically grow, so the reason of different results between WAZ and WHZ is not clear to us so far. This study provided just a basis for monitoring and evaluating currently implemented interventions, which is a limitation of it”. (lines 279-283).
- A quick google search brought up this study that also assessed the impacts of the intervention using data collected during the intervention rather than DHS data (https://onlinelibrary.wiley.com/doi/full/10.1111/mcn.13630). It seems like it is also important to discuss your findings in the context of these findings, which differ (and any other findings related to the impacts of this program that have been published).
Response:
Thank you very much for your comments.
This paper seems to have been published while we’ve been brushing our draft for the journal. So, it is not referred to in the draft.
This paper differs from the following points compared with our paper:
- Their research used primary data based on 4 districts out of 34 in the comparison group and 4 districts out of 41 in the treatment group, which is a small sample.
On the other hand, we used all possible data (from demographic health survey) as a comparison group and a treatment group.
- Their research employed a big distance difference between baseline survey (2012) and end line survey (2022) (around 10 years).
We’re afraid many factors may affect during this long period such as changed societal habits, knowledge or perceptions, changes in technology, public policy, economic conditions, and health care system.
Our research focused on Suaahara I (2011-2016) only, which can be sharply evaluated about the impact.
- Their research hasn’t done the pre-trend test to check the similarity of the control group and the treatment group before the intervention, which is insufficient to justify the assumption that both groups have similarities.
Our research has done the pre-trend test to check the characteristics of both groups.
- Lines 292-293: These are very sweeping claims. Please limit claims to those that can actually be supported by your findings. The only impact on morbidity was on the prevalence of fever, and you did not show any impacts on mortality.
Response:
Thank you for your comment.
We have deleted the above claims (from line 368 to 374).
- Lines 296-398: Again, please limit claims to those actually evaluated by the study. You did not assess the impact of the intervention on healthcare costs, women’s SES, or women’s empowerment.
Response:
Thank you for your comment.
We have deleted the claims mentioned above, as we did in the previous response.
- Line 299: Again, you did not assess mortality rates. This entire paragraph is not supported by your findings.
Response:
Thank you for your comment.
We have deleted the claims mentioned above, as we did in the previous response.
- Lines 315-330: This section is not really conclusions but rather a line-by-line restatement of the findings. To me, an effective conclusions section should provide a concise summary of the key findings and, importantly, why those findings are important and how they can help inform policy.
Response:
Thank you for your comment.
This research shows the impact of Suaahara interventions on the nutritional recovery of children under 2 years. Nutrition education, awareness, and income-generating activity in this program improve nutritional practices for women and children. The most valuable resource for any nation is its children, whose physical, mental, and emotional growth will be vital for them as they grow up and become citizens. The health of the children is greatly influenced by the mothers' knowledge of child nutrition (Saurish Hegde et al.).
Saurish Hegde, Praveen Kulkarni, Jay Gohri, Mayuri Chaurasia, R. Pragadesh, Aisha Siddiqua, K. Shreyaswini, and KS Sahana, Effectiveness of nutritional awareness in child diet among mothers of under five in anganwadi in Mysuru: A quasi-experimental study.
So, we added some interpretation and implications to the last part of the conclusion part as follows:
“Above these, we can summarize that this program could improve the health and nutrition condition of both infants and their mothers significantly around 6% to 10% of health indicators in Nepal. It shows that this kind of health care policy in low- and middle-income countries is crucial for human resources development. Nutrition education, awareness, and income-generating activity in this program improve nutritional practices for women and children. The most valuable resource for any nation is its children, whose physical, mental, and emotional growth will be vital for them as they grow up and become citizens. The health of the children is greatly influenced by the mothers' knowledge of child nutrition [22].
At least, if they introduce the healthcare policy, the evaluation of the policy on how much it affected objectives is necessary for policymakers. Clarifying whether public policy or program is effective or not is very important for any country facing a budget constraint now and in the future. Also, this process is essential for policymakers to propose evidence-based policymaking.”

Round 2
Reviewer 1 Report
Comments and Suggestions for Authors
Please revise the long complicated sentence on L36-38
Table 1 and 5 on Lines 216 and 258: there is no p-values of <0.05 or p<0.1 stated in the Table. Why are you mentioning these are legends?
What about the ethical considerations? From where were the ethical approvals obtained
Comments on the Quality of English LanguageMinor amendments
Author Response
- Please revise the long complicated sentence on L36-38
Response:
Thank you for this comment.
We revised this sentence as follows:
“Investments in nutrition during the first 1000 days from pregnancy to a child's second birthday are very crucial for babies [1,2].” (lines 36-37)
- Table 1 and 5 on Lines 216 and 258: there is no p-values of <0.05 or p < 0.1 stated in the Table. Why are you mentioning these are legends?
Response:
Thank you for this comment.
We dropped those unnecessary p-value legends.
- What about the ethical considerations? From where were the ethical approvals obtained
Response:
Thank you for your asking.
This Suaahara program was implemented by the central and local governments, and we used those secondary data so they didn’t have to obtain ethical approvals.

Reviewer 2 Report
Comments and Suggestions for Authors
I thank the authors for their revisions to the paper and responses to my previous round of comments. I have the following remaining comments. Most are minor, but I really do think the authors need to add another table to the results section that summarizes the main outcome variables by DHS round and treatment group. My comments are:
· Title: I have no idea what “mass health indicators” means. I think the title needs to be revisited again. How about “Assessment of a community-based nutrition program for women and children in Nepal using DHS data” or something like that?
· Abstract line 21: This should be “evidence base on the impact” rather than "evidence based on the impact" – that is, your findings are part of the body of evidence (i.e., the evidence base) on the impact of the program.
· Abstract lines 25-26: You edit, “…during 5 years before the program in 2011 and after 25 the program in 2016” doesn’t really make sense as written. Can you please rephrase for clarity?
· In my previous review, I asked that you “…please also simply present your outcomes for children and PLW by DHS round and treatment group (i.e., mean HAZ, WAZ, WHZ, ANC visits, fever, diarrhea, water and sanitation outcomes, etc).” You have not addressed this comment (your response is related to Table 2 and the beta coefficients, which doesn’t provide any sense of the actual value of those variables before and after the intervention). I want to know the average HAZ, WAZ, number of ANC visits, etc by DHS round and treatment group. Otherwise, it’s is not possible to really interpret the size of the estimated impacts. To address this comment, you need to add a new table to the results section.
· Lines 328-329: It is not clear what is meant by “…significantly around 6 % to 10% of health indicators in Nepal.” What is the 6-10% referring to?
Comments on the Quality of English LanguageSome sentences contain unclear phrasing that should be addressed, but in general the quality of the English Language is acceptable.
Author Response
- I thank the authors for their revisions to the paper and responses to my previous round of comments. I have the following remaining comments. Most are minor, but I really do think the authors need to add another table to the results section that summarizes the main outcome variables by DHS round and treatment group. My comments are:
Response:
Thank you for your comments.
We added a summary table about the average difference between the two groups and the figure showing their parallel trend in the result part and appendix.
- Title: I have no idea what “mass health indicators” means. I think the title needs to be revisited again. How about “Assessment of a community-based nutrition program for women and children in Nepal using DHS data” or something like that?
Response:
Thank you for your suggestion.
We revised the title to “Assessment of a Community-Based Nutrition Program for Women and Children in Nepal Using Demographic and Health Survey”.
- Abstract line 21: This should be “evidence base on the impact” rather than "evidence based on the impact" – that is, your findings are part of the body of evidence (i.e., the evidence base) on the impact of the program.
Response:
Thank you for your comments. We changed that word to “evidence on the impact”, which seems to be clear.
- Abstract lines 25-26: You edit, “…during 5 years before the program in 2011 and after 25 the program in 2016” doesn’t really make sense as written. Can you please rephrase for clarity?
Response:
Thank you for your comments. We revised to clarify the meaning as follows:
“during 5 years before and after the program”. (line 26)
- In my previous review, I asked that you “…please also simply present your outcomes for children and PLW by DHS round and treatment group (i.e., mean HAZ, WAZ, WHZ, ANC visits, fever, diarrhea, water and sanitation outcomes, etc.).” You have not addressed this comment (your response is related to Table 2 and the beta coefficients, which doesn’t provide any sense of the actual value of those variables before and after the intervention). I want to know the average HAZ, WAZ, number of ANC visits, etc. by DHS round and treatment group. Otherwise, it’s not possible to really interpret the size of the estimated impacts. To address this comment, you need to add a new table to the results section.
Response:
Thank you for your comments.
We added a summary table about the average difference between the two groups. Also, to check those average trends before and after the program, we added another figure that shows the parallel trend check in the appendix.
We believe that it’s sufficiently possible to interpret the impact of program with these.
- Lines 328-329: It is not clear what is meant by “…significantly around 6% to 10% of health indicators in Nepal.” What is the 6-10% referring to?
Response:
Thank you for your comments. We revised this sentence as follows:
“we can summarize that this program could improve the health condition of both infants and their mothers significantly in some facts like ANC visits, safe deliveries, the prevalence of fever in children, and so on in Nepal.” (lines 326-328)
